# COVID-19 and mental health in 8 low- and middle-income countries: A prospective cohort study

**Nursena Aksunger** [1‡], **Corey Vernot** [2‡], **Rebecca Littman** [3‡], **Maarten Voors** [4‡], **Niccolò F. Meriggi** [5‡], **Amanuel Abajobir** [6], **Bernd Beber** [7], **Katherine Dai** [8], **Dennis Egger** [9], **Asad Islam** [10,11], **Jocelyn Kelly** [12], **Arjun Kharel** [13,14], **Amani Matabaro** [15], **Andrés Moya** [16], **Pheliciah Mwachofi** [17], **Carolyn Nekesa** [18], **Eric Ochieng** [19], **Tabassum Rahman** [20], **Alexandra Scacco** [21], **Yvonne van Dalen** [22], **Michael Walker** [23], **Wendy Janssens** [1‡], **Ahmed Mushfiq Mobarak** [24,25‡]*

1 School of Business and Economics, Vrije Universiteit Amsterdam, Amsterdam, the Netherlands, 2 Y-RISE, New Haven, Connecticut, United States of America, 3 Department of Psychology, University of Illinois at Chicago, Chicago, Illinois, United States of America, 4 Department of Social Sciences, Wageningen University, Wageningen, the Netherlands, 5 International Growth Centre, Freetown, Sierra Leone, 6 African Population and Health Research Center, Nairobi, Kenya, 7 RWI—Leibniz Institute for Economic Research, Berlin, Germany, 8 Department of Economics, New Haven, Connecticut, United States of America, 9 Department of Economics, University of California, Berkeley, California, United States of America, 10 Department of Economics, Monash University, Melbourne, Victoria, Australia, 11 J-PAL, Cambridge, Massachusetts, United States of America, 12 Harvard Humanitarian Initiative, Cambridge, Massachusetts, United States of America, 13 Department of Sociology, Tribhuvan University, Kathmandu, Nepal, 14 Centre for the Study of Labour and Mobility, Kathmandu, Nepal, 15 Action Kivu, Bukavu, Democratic Republic of Congo, 16 School of Economics, Universidad de los Andes, Bogotá, Colombia, 17 Innovations for Poverty Action, Abuja, Nigeria, 18 Vyxer Research Management and Information Technology (REMIT), Busia, Kenya, 19 Innovations for Poverty Action, Nairobi, Kenya, 20 Centre for Epidemiology and Biostatistics, University of Melbourne, Melbourne, Victoria, Australia, 21 WZB Berlin Social Science Center, Berlin, Germany, 22 100 Weeks, Amsterdam, the Netherlands, 23 Center for Effective Global Action, University of California, Berkeley, California, United States of America, 24 School of Management, Yale University, New Haven, Connecticut, United States of America, 25 Deakin Business School, Deakin University, Melbourne, Victoria, Australia

‡ These authors share first authorship on this work. WJ and AMM share last authorship on this work.
* ahmed.mobarak@yale.edu

**Data Availability Statement:** All replication files (de-identified data and code) are deposited in Harvard Dataverse. The replication file for the paper

## Abstract

### Background

The Coronavirus Disease 2019 (COVID-19) pandemic and associated mitigation policies created a global economic and health crisis of unprecedented depth and scale, raising the estimated prevalence of depression by more than a quarter in high-income countries. Low- and middle-income countries (LMICs) suffered the negative effects on living standards the most severely. However, the consequences of the pandemic for mental health in LMICs have received less attention. Therefore, this study assesses the association between the COVID-19 crisis and mental health in 8 LMICs.

### Methods and findings

We conducted a prospective cohort study to examine the correlation between the COVID-19 pandemic and mental health in 10 populations from 8 LMICs in Asia, Africa, and South

is available via this link https://doi.org/10.7910/DVN/RFODZC.

**Funding:** This work was supported by Saving Brains–Grand Challenges Canada (SB-POC-1809-19901 to A.M., https://www.grandchallenges.ca/programs/saving-brains/); United Way Colombia (https://unitedwaycolombia.org/), Fundación Éxito (https://www.fundacionexito.org/), Fundación FEMSA (https://fundacionfemsa.org/), Genesis Foundation, Fundación Coca-Cola (https://www.fundacioncocacola.com/), Primero lo Primero (https://primeroloprimero.co/es/), Universidad de los Andes (https://uniandes.edu.co/); Dutch Research Council (VI.Vidi.191.154 to M.V., https://www.nwo.nl/); NORC at the University of Chicago (to M.V., https://www.norc.org/Pages/default.aspx); Joep Lange Institute (16801.03 for W.J. and A.A., www.joeplangeinstitute.org); Sint Antonius Stichting (1913112 to Y.v.D. ,https://www.aham.nl/sintantoniusstichtingprojecten/); BRAC (to A.I., http://www.brac.net/) and LEGO Foundation (to A.I., https://learningthroughplay.com/); the UK Foreign, Commonwealth & Development Office (awarded through Innovation for Poverty Action's Peace & Recovery Program to B.B., A.S., P.M.); IPA Peace and Recovery Initiative (to M.W.), IZA G2LMLIC (to M.W.), Berkeley Population Center (to M.W.), NIH (to M.W.) and GiveWell (to M.W.); International Growth Centre (to D.E. and M.W.), PEDL (to D.E. and M.W.), NSF (to D.E. and M.W.) and Weiss Family Fund (to D.E. and M.W.). The funders had no role in study design, data collection and analysis, decision to publish, or preparation of the manuscript.

**Competing interests:** The authors have declared that no competing interests exist.

**Abbreviations:** CES-D, Center for Epidemiological Studies-Depression; CI, confidence interval; COVID-19, Coronavirus Disease 2019; HIC, high-income country; IRB, Institutional Review Board; LMIC, low- and middle-income country; RD, regression discontinuity; SCL-90R, Symptom Checklist-90-R; SD, standard deviation; SMD, standardized mean difference; SSA, sub-Saharan Africa; WHO-5, World Health Organization Well-Being Index 5; YLD, years of healthy life lost due to disability.

America. The analysis included 21,162 individuals (mean age 38.01 years, 64% female) who were interviewed at least once pre- as well as post-pandemic. The total number of survey waves ranged from 2 to 17 (mean 7.1). Our individual-level primary outcome measure was based on validated screening tools for depression and a weighted index of depression questions, dependent on the sample. Sample-specific estimates and 95% confidence intervals (CIs) for the association between COVID-19 periods and mental health were estimated using linear regressions with individual fixed effects, controlling for independent time trends and seasonal variation in mental health where possible. In addition, a regression discontinuity design was used for the samples with multiple surveys conducted just before and after the onset of the pandemic. We aggregated sample-specific coefficients using a random-effects model, distinguishing between estimates for the short (0 to 4 months) and longer term (4+ months). The random-effects aggregation showed that depression symptoms are associated with a increase by 0.29 standard deviations (SDs) (95% CI [−.47, −.11], $p$-value = 0.002) in the 4 months following the onset of the pandemic. This change was equivalent to moving from the 50th to the 63rd percentile in our median sample. Although aggregate depression is correlated with a decline to 0.21 SD (95% CI [−0.07, −.34], $p$-value = 0.003) in the period thereafter, the average recovery of 0.07 SD (95% CI [−0.09, .22], $p$-value = 0.41) was not statistically significant. The observed trends were consistent across countries and robust to alternative specifications. Two limitations of our study are that not all samples are representative of the national population, and the mental health measures differ across samples.

## Conclusions

Controlling for seasonality, we documented a large, significant, negative association of the pandemic on mental health, especially during the early months of lockdown. The magnitude is comparable (but opposite) to the effects of cash transfers and multifaceted antipoverty programs on mental health in LMICs. Absent policy interventions, the pandemic could be associated with a lasting legacy of depression, particularly in settings with limited mental health support services, such as in many LMICs. We also demonstrated that mental health fluctuates with agricultural crop cycles, deteriorating during "lean", pre-harvest periods and recovering thereafter. Ignoring such seasonal variations in mental health may lead to unreliable inferences about the association between the pandemic and mental health.

## Author summary

### Why was this study done?

- The worldwide economic and health crises triggered by the Coronavirus Disease 2019 (COVID-19) pandemic have had a significant influence on mental health, with the estimated prevalence of depression having increased by more than 25% in high-income countries.

- Although the adverse consequences of the pandemic on living standards have been most severe in low- and middle-income countries (LMICs), the consequences of the pandemic for mental health in LMICs have received less attention.

## What did the researchers do and find?

- The purpose of this research is to investigate the association between the COVID-19 pandemic and mental health in 8 LMICs in Asia, Africa, and South America.

- Before and during the pandemic, the mental health of 21,162 individuals (mean age 38.01 years, 64.0% female) was measured using survey data.

- Our individual-level primary outcome measure was based on validated depression screening instruments and a sample-specific weighted index of depression questions.

- We found that depression symptoms were associated with a significant increase in the 4 months following the onset of the pandemic (0.29 standard deviations (SDs), 95% confidence interval (CI) [−.47, −.11], $p$-value = 0.002) and that the average recovery of 0.07 SD was not statistically significant in the subsequent period (95% CI [−0.09, .22], $p$-value = 0.41).

## What do these findings mean?

- We showed a substantial negative correlation between the COVID-19 pandemic and mental health after adjusting for seasonality, suggesting that the pandemic might induce long-term depression, especially in LMICs with poor mental health support facilities.

- We also provided evidence for seasonal changes in mental health depending on agricultural crop cycle. This seasonality should be considered when examining changes in mental health over time in order to prevent drawing inaccurate conclusions.

- The observed trends were consistent across countries and robust to alternative analyses, although the study was limited by the fact that not all samples were representative of the national population and the mental health indicators differed among samples.

## Introduction

The Coronavirus Disease 2019 (COVID-19) pandemic and associated containment policies sparked a global economic crisis of unprecedented depth and scale. The adverse effects on living standards were most acutely felt in low- and middle-income countries (LMICs) [1,2]. A global review primarily focused on studies in high-income countries (HICs) reports that the combination of the COVID-19 death toll and the economic downturn increased the prevalence of depression by 27.6% [3]. The virus sparked widespread fear of infection. The stigma of infection generated anxiety, deaths of family members caused anguish, loss of employment and income created economic stress and food insecurity, and mobility restrictions and lockdowns caused separation, loneliness, mental distress, and higher suicidal tendency [4–7]. While the available evidence documenting these effects has focused more on HICs [8–17], these phenomena may be more severe in LMICs. Many LMICs experienced strict containment measures and suffer from poorer access to healthcare, low health insurance coverage, shortage of medical supplies (including COVID-19 vaccines), informal labor markets characterized by a lack of social safety nets, and uncertainties created by a lack of relevant information [18].

Due to the limited availability of data, the consequences of the COVID-19 pandemic in LMICs have received less attention in the academic literature and policy dialogues. Limited

pre-COVID-19 period data also make it more challenging to track changes in the prevalence of mental health problems in LMICs post-pandemic [19,20]. During the pandemic, stress may have been more acute in LMIC populations due to more profound economic crises that resulted in widespread food insecurity and underlying risk factors for mental health [2]. An increase in the prevalence of depression could also be more consequential for LMIC societies due to the lack of resources devoted to mental health services and the potential impairment in economic productivity [21]. While 1 billion people are currently suffering from mental or neurological disorders worldwide [22], 82% of people living with a mental health disorder reside in LMICs [22,23]. LMIC residents account for more than 80% of the years of healthy life lost due to disability (YLD) associated with mental disorders [23]. Although there are known effective treatments, mental health is underresourced, with less than US$2 annual spending per person globally and US$0.25 per person in LMICs [24]. Still, less than 1.6% of national health budgets are spent on mental health in these countries, with only one psychiatrist serving 200,000+ people on average [22,25]. More than 75% of people with mental disorders in LMICs never seek treatment from professionals [26]. These statistics show a high disease burden overlaid with low mental healthcare coverage, which may impede escaping from poverty through psychological poverty traps. Several studies point out that while poverty is a risk factor for mental illnesses, untreated mental health problems, in turn, may deepen poverty through worse cognitive functioning and short-sighted decision-making [27,28] and may hinder income generation. Therefore, the adverse effects of the pandemic may persist well after the COVID-19 situation is controlled, affecting the well-being of future generations, especially in LMICs.

This study assesses the association between depression, the most prevalent mental disorder among adults, and the COVID-19 pandemic in LMICs. While a few studies have analyzed how mental health changed before and after COVID-19 in individual LMICs [29,30], our study is distinctive in that we track populations over multiple rounds, both pre-and post-pandemic, in 8 countries from sub-Saharan Africa (SSA), South Asia, and South America. Crucially, this allows us to explicitly adjust for time trends and to take into account the independent effect of seasonal fluctuations in mental health related to lean and affluent periods driven by local crop and harvest cycles. In 2 countries, we collected high-frequency data on mental health that we can directly match to seasonal poverty and hunger, allowing us to investigate the importance of this adjustment in detail.

## Methods

### Study design and participants

We performed a prospective observational cohort study in 10 different samples across 8 countries—Bangladesh, Colombia, Democratic Republic of the Congo, Kenya, Nepal, Nigeria, Rwanda, and Sierra Leone (denoted BGD, COL, DRC, KEN1, KEN2, KEN3, NPL, NGA, RWA, and SLE in S1 Table)—that had a combined population of nearly 525 million. All our samples were from specific subpopulations within each country. Descriptive statistics for each sample are provided in the S2 Table.

Each sample included an assessment of depression symptoms collected before and after the onset of the COVID-19 pandemic. For our main analysis, we denoted the "onset" as the date when COVID-19 was declared as a pandemic by the World Health Organization (11 March 2020). Because stringency measures for the containment of COVID-19 had been implemented at different times in different countries, in our analysis, we also varied the date of the COVID-19 onset by country and defined this as the date of the largest increase in COVID-19 mitigation policy stringency. Information on the exact timing of the first COVID-19 case,

government-imposed social distancing policies, and survey implementation dates for all samples are described in **S1 Appendix** in the Supporting information.

Data were collected from the same households in each survey wave. An advantage of this panel structure was that we had pre-COVID-19 baseline data on mental health for all samples. The primary reporting unit for the mental health survey questions was the individual. The total dataset comprised 68,064 interviews encompassing 21,162 unique individuals with the number of surveys ranging from 2 to 17 (mean 7.1).

## Survey methods and timing

In most samples, post-COVID-19 data were collected via telephone interviews to minimize in-person contact and comply with government social distancing guidelines. Interviews were conducted by local enumerators in each country, with procedures to match languages, dialects, and accents between respondents and enumerators. Phone surveys enabled rapid data collection. However, a drawback of the switch to phone-based data collection was that it may lead to selective attrition of low-income families without phones or living in low-connectivity areas. In one sample (KEN3), this was handled by providing a basic feature phone to respondents who did not have one. For the DRC sample, phone surveys were impossible due to the low coverage of mobile phone networks, and data were collected through in-person interviews.

The COVID-19 pandemic and mental health outcomes may show a spurious relationship depending on the calendar time at data collection. In **S1 Fig**, we summarize the pre- and post-COVID-19 survey dates in each sample and relate them to the timing of the lean season and onset of the COVID-19 pandemic. The post-COVID-19 survey waves in 7 samples (BGD, KEN1, KEN2, KEN3, NPL, NGA, and RWA) overlapped with the "lean" season when food stocks in the rural population are generally low, and food prices are high. While this decreased households' capacity to cope with economic shocks, it was also a period of low agricultural labor demand in these settings. This may have had independent effects on household resilience to unemployment and mental health shocks [2]. For example, at the outbreak of the pandemic in March 2020, many rural households in Bangladesh, the Democratic Republic of the Congo, and Rwanda were entering the "lean" season. Three samples (COL, KEN3, and RWA) conducted high-frequency surveys spanning a long-enough period to examine the longer-term effect of the COVID-19 pandemic. In 2 samples (NPL and KEN2), we could directly compare mental health changes with the typical food security patterns reported in previous years, as described in more detail below.

## Construction of main variables

Our primary outcome was an index of symptoms related to depression. We used 2 different methods to calculate the depression index, dependent on data availability. For the 5 samples that collected data using a validated mental health screening tool (BGD, COL, NGA, DRC, and KEN3), we followed the prescribed calculation method, i.e., the summed scores of the Center for Epidemiological Studies-Depression (CES-D) scale for BGD and KEN3, the Symptom Checklist-90-R (SCL-90R) in COL, the 15-item Hopkins Symptom Checklist for DRC, and the World Health Organization Well-Being Index 5 (WHO-5) for NGA. We then reverse-scored and transformed these measures to have a standard deviation (SD) equal to 1 across individuals in the first baseline wave of our panel data, with higher scores showing better mental health.

In the remaining samples (KEN1, KEN2, SLE, NPL, RWA), only part of the questions used to measure mental health came from a standardized tool. In those samples, we used only those depression-related questions that were common across all surveys. To maximize consistency

in reporting results, we constructed indices of the available depression items as follows. First, we standardized the ordinal scales of each depression item to have a mean 0 and an SD 1 in the baseline survey. We then constructed the index as the weighted average of these standardized depression items, using 3 different weighting methods. **S3 Table** in the Supporting information provides detailed descriptions of how the mental health outcome variables were constructed in each sample. The reference period in all questionnaires was the previous 7 days unless otherwise noted.

We constructed 3 measures of mental health, which all used the scores based on validated tools where available, but which differed in the weighting method for the samples without validated scores. Our preferred measure weighed all items equally. The second measure used the first factor loadings from a factor analysis as weights. This measure placed more weight on strongly correlated items with the goal of identifying the single most important latent variable. The third measure used an inverse-covariance weighted index [31]. This measure placed less weight on strongly correlated items to capture variation in many potential latent variables.

In **S2 Fig**, we also plot levels of the depression index over time and include the "COVID-19 case counts" and a measure of the "COVID-19 stringency index" for reference. Data were retrieved from WHO and ourworldindata.org data sources, respectively. The COVID-19 stringency index was a composite metric based on 9 categorical policy indicators, such as school closures, workplace closures, and travel bans, rescaled to range from 0 to 100 (100 being the most stringent) [32]. We used the stringency index at the national level for all samples.

For 2 samples where we conducted a more intensive examination of seasonality, we created a food insecurity index by summing standardized items related to whether one or more household members skipped meals or reduced portion size or diet quality in the past 7 days (KEN2) or 14 days (NPL). These indices were constructed similarly to our mental health index described above, where we summed standardized (in SD units) ordinal integer-valued responses to food insecurity questions.

## Empirical methodology

A straightforward and commonly used strategy to investigate the mental health effects of COVID-19 would be to compare average mental health outcomes for the same individuals at pre- and post-pandemic moments. An implicit assumption in such comparisons is that average mental health levels are not changing over time in the absence of the pandemic. However, there is evidence that mental health indicators fluctuate with agricultural seasons and other events affecting the broader economy [27,33,34]. In our own data, we saw fluctuating pre-COVID-19 trends in mental health in 5 of the 6 samples for which we had multiple pre-COVID-19 survey waves. To assess changes due to the COVID-19 pandemic, we therefore preferred strategies that adjust for non-COVID-19-related trends in mental health, as much as each dataset allowed. We used linear regressions with individual fixed effects for sample-specific estimates, as well as a regression discontinuity (RD) design for the samples with high-frequency data just prior to and after the onset of COVID-19.

Some, but not all, of our samples had data that allowed us to adjust for seasonal non-COVID-19 trends directly. Wherever possible, we made these adjustments. In samples where we did not have the data necessary to make adjustments directly, we estimated simpler models with the best possible indirect controls for trends in mental health status unrelated to the pandemic. So, for each sample separately, we compared depression levels in a given month post-COVID-19 with depression levels in that same month pre-COVID-19 (if available) plus any year-on-year trend that we estimated from our year-on-year data (if available). The sample-specific strategies are described in detail below. This approach implied that our analysis

combined different estimation strategies for different samples depending on the data structure in each sample.

An alternative would have been to ignore non-COVID-19 trends altogether and estimate the simpler model without adjustments in all samples. While this alternative would have had the benefits of simplicity and consistency, we preferred estimates that used what we believed was the best estimation strategy in each sample, given the scope of the data we had. Nevertheless, we also report results without these adjustments in **S3 Fig**.

Although we used different preferred models in different samples, we still estimated a single average effect of the pandemic by combining our estimates from different samples as one would in a meta-analysis. We aggregated sample-specific coefficients using a random-effects model, the standard tool for meta-analysis, where we allowed the true effect to vary across different studies.

## Estimation strategies per sample

We controlled for both month and year trends in our estimation in the most extended samples with multiple survey waves pre-COVID-19 spanning a full year or more. Specifically, we estimated:

$$Y_{i,t,s} = \alpha_{i,s} + \beta_{t,s} + \delta_{s,month(t)} + \kappa_s year_t + \epsilon_{i,t,s} \tag{1}$$

where $Y_{i,t,s}$ was the mental health outcome of individual $i$ at time $t$ in sample $s$, and $\alpha_{i,s}$ captured individual fixed effects. $t = 0$ during the pre-COVID period, taking on integer values for discrete time periods in the post-COVID period. The entire pre-COVID-19 period was our omitted category in all specifications ($\beta_{0,s} = 0$), implying that $\beta_{t,s}$ was the difference between mental health during the pre-COVID period and period $t$ in sample $s$. We reported heteroskedasticity-robust standard errors that also allowed for arbitrary correlation between responses by the same participant using an Eicker–Huber–White "sandwich" covariance estimator [35].

Depending on the granularity of $t$ in a particular sample, we could observe multiple post-COVID-19 time periods, allowing for different correlation with mental health earlier versus later in the pandemic. When we aggregated results across samples, we divided the post-COVID-19 period into 2 time periods: 0 to 4 months post-COVID-19 and more than 4 months post-COVID-19. We chose this relatively small number of post-COVID-19 time periods to allow us to aggregate across many samples with different timings of surveys, while still allowing for some dynamics in the correlation with mental health throughout the pandemic. When estimating models for individual samples, we used more granular time periods post-COVID-19, defining time periods of 0 to 2, 2 to 4, 4 to 6, 6 to 9, 9 to 12, and 12 to 15 months post-COVID-19.

In our samples with multiple years of pre-COVID-19 data (RWA, COL, KEN1), we allowed for both seasonal trends and year-on-year trends in mental health in our model as in Eq (1). Here, $\delta_{s,month(t)}$ was a month-of-year fixed-effect and $\kappa_s$ was the year-on-year increase in mental health prior to the pandemic.

In our samples with multiple waves of pre-COVID-19 data spanning less than a full calendar year, but with supplemental data on typical seasonal food security (KEN2 and NPL), we estimated Eq (2), in which we controlled for the average level of the food insecurity index in respective months pre-COVID-19, designated as $x_{t,s}$, as follows:

$$Y_{i,t,s} = \alpha_{i,s} + \beta_{t,s} + \gamma x_{t,s} + \epsilon_{i,t,s} \tag{2}$$

where $\beta_{t,s}$ was zero during all pre-COVID-19 time periods, as in Eq (1).

In the KEN2 sample, pre-COVID-19 mental health data were collected 3 years prior to the pandemic. This large gap in time between the pre- and post-COVID-19 data collection could exacerbate bias from secular trends in mental health over time. If mental health was changing substantially from year to year, the more time in-between our pre-COVID-19 data and the pandemic, the worse of a counterfactual that pre-data would be. However, the KEN2 sample collected data on food security over full calendar years that spanned both 2014 to 2015 and 2016 to 2017. This allowed us to estimate and incorporate any secular trends in food security into our control variables. This is reflected in Eq (3):

$$Y_{i,t,s} = \alpha_{i,s} + \beta_{t,s} + \gamma \hat{x}_{t,s} + \epsilon_{i,t,s} \tag{3}$$

where $\hat{x}_{t,s}$ was estimated from the 2014 to 2015 and 2016 to 2017 data on food security with both month-of-year and linear year-on-year time trends, as in Eq (4):

$$\hat{x}_{t,s} = \delta_{s,month(t)} + \kappa_s year_{i,t,s} \tag{4}$$

Here, $year_{i,t,s}$ was a continuous variable indicating the date the survey was conducted and transformed into yearly units for interpretation. During the pre-COVID-19 period, the value was set to the exact date for each survey, so that we could use this variation to estimate $\kappa_s$. During the post-COVID-19 period, the value of $year_{i,t,s}$ was set to the average survey date of all respondents in discrete time period $t$ in sample $s$. This was done intentionally so that $year_{i,t,s}$ would be collinear with our indicators for discrete time periods during the post-COVID-19 period, and only variation from the pre-COVID-19 period would be used to estimate $\kappa_s$.

For one sample (KEN3), we had multiple waves of pre-COVID-19 data spanning less than a full year, and we did not have supplemental data on typical seasonal food security. We chose to control for linear time trends in this sample. In the remaining samples with only one wave of pre-COVID-19 data (BGD, DRC, NGA, SLE), we regressed mental health on individual fixed-effects and the pre-post-COVID-19 time period categories only.

To estimate the average effect of the pandemic on mental health over all samples, we aggregated the estimates across countries using a random-effects model and estimated the intersample variance in effects $\tau^2$ using restricted maximum likelihood (following recent simulation studies; [36–38]).

For figures showing trends in mental health for individual samples, we split post-COVID-19 survey waves into 1-month intervals and plotted the average survey date during the time period on the x-axis. In some cases where very few interviews were conducted in the final month of a survey wave and the interview dates did not span the entire final month, we merged the final 2 months of the survey wave into one time period that spanned between 1 and 2 months.

Our specifications controlling for time trends relied on the assumption that pre-pandemic trends continued linearly during the post-pandemic period. This assumption may have been too strong, especially when we used a relatively short period of pre-COVID-19 data to estimate the counterfactual mental health status long into the pandemic (as was the case for the KEN2 sample). Instead, a weaker assumption was that independent time trends in mental health outcomes were locally linear around the onset of the pandemic. With this assumption, we could estimate the very short-run association of the pandemic by comparing mental health outcomes just before and just after the outbreak. This approach of comparing outcomes before and after a specific threshold is known as a regression discontinuity (RD) design. For a survey of RD applications in economics, see [39]. We estimated the RD specification around the initial onset of COVID-19 in the 5 samples where we had multiple pre- and post-COVID-19 survey waves

within 3 months of the start of the pandemic (COL, KEN1, KEN3, NPL, RWA), and we calculated the average RD-effect using a random-effects model as described above.

Statistical analyses were conducted using Stata (version 15). All statistical tests were two-sided, and the threshold for statistical significance was set at $p < 0.05$. This study was reported as per the Strengthening the Reporting of Observational Studies in Epidemiology (STROBE) guidelines **S2 Appendix**.

We used an internal analysis plan to help instruct each sample team what data to deliver and in what format, how we would define our outcome variable, what analysis we would run and any restrictions. We discuss the original analysis plan and modifications in **S3 Appendix** in the Supporting information.

## Statement of ethics

The studies BGD, COL, DRC, KEN1, KEN2, KEN3, NPL, NGA, RWA, and SLE received Institutional Review Board (IRB) approval from, respectively, BRAC University (ref no. 2019-028-ER); the Ethics Committee of the Universidad de los Andes, Colombia (protocol 786, 2017); the NORC Institutional Review Board (IRB00000967); UC Berkeley, Maseno University; AMREF Ethics and Scientific Review Committee Kenya (P679/2019); Yale University (IRB Protocol 2000025621); IPA (IRB Protocol 15057); Vrije Universiteit Amsterdam (20210413.1); and Sierra Leone Ethics and Scientific Review Committee (SLERC 2904202) and Wageningen University (24062020). Informed consent was sought from participants prior to any survey activities. Participants in each sample gave their verbal informed consent before each survey took place. Each sample team minimized the possibility of coercion or influence, and the respondents were given sufficient time to consider participation. Additional information regarding the ethical, cultural, and scientific considerations specific to inclusivity in global research is included in the **S4 Appendix**.

## Results

### Seasonal trends in mental health

We started with an investigation of seasonal trends in depression. **Fig 1** zooms in on 2 of our samples (NPL and KEN2), for which we observed mental health across multiple years. We had less than a full year of pre-COVID-19 data in these samples, so we could not adjust for independent year-on-year improvements in mental health. However, we did have detailed pre-COVID-19 data on typical trends in seasonal food insecurity in these samples, which we could use to adjust for recurring agricultural seasons.

**Fig 1** shows the evolution of seasonal food security (indicated by the continuous blue line) and mental health (indicated by red points and error bars) over time before and after the onset of the pandemic (indicated by the red vertical dashed line). The line for food security is based on the predicted values from a lowess model using data from an index of food security items. Fig 1A shows data from our NPL sample. Fig 1B shows data from our KEN2 sample. In Fig 1B, the solid blue line shows the 2016 to 2017 period during which we had both mental health data and food security data that we used to estimate the lowess. The dashed line is our extrapolated prediction based on this data. Not pictured are the 2014 to 2015 period during which we had food security data but no mental health data. The red dots and vertical lines are predicted values and associated 95% confidence intervals (CIs) for our positively coded depression index. The predicted values were based on discrete 1-month intervals using coefficients from Eq (1). The x-axis value for each dot and CI is the average date of surveys conducted during that time interval.

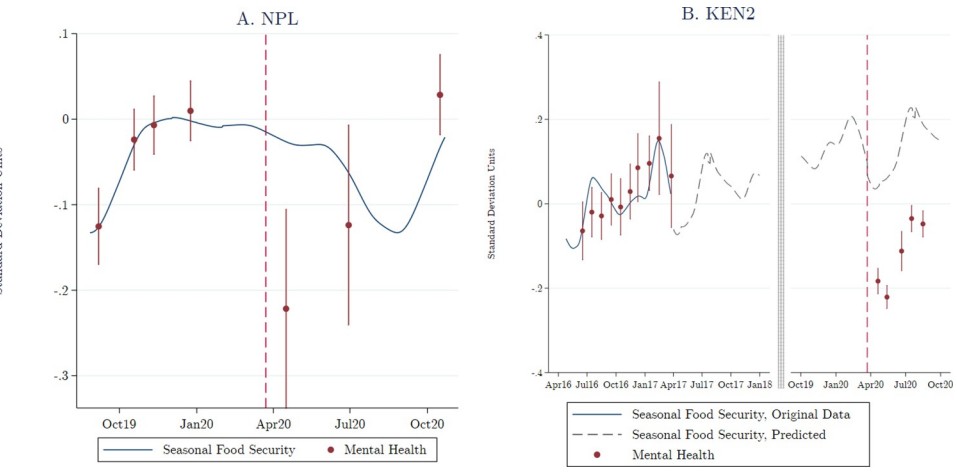

**Fig 1. Seasonal food security and pre-trends.**

In both samples, we saw clear evidence of a positive trend in mental health status prior to the pandemic, closely following the fluctuations in food security (comparing the blue line and the red dots in **Fig 1**). We also saw that mental health in April 2020, immediately after the onset of the pandemic, was well below the pre-COVID-19 average and improved substantially from April to October 2020. However, it is difficult to interpret these post-COVID-19 fluctuations without further information on seasonal trends.

To paint a clearer picture of the mental health consequences of COVID-19, we therefore added data on typical seasonal trends in food security. In NPL, the "lean" season occurred roughly from July to September, and we saw little difference between food security in April and October of a typical year. This suggests that the pandemic had an immediate negative effect on mental health in April, which faded over time. In contrast, in KEN2, food security rose substantially in a typical year from April to October. Moreover, the 2014 to 2015 and 2016 to 2017 data on food security showed an independent average increase of 0.06 SD per year in the food security index. We therefore incorporated this year-on-year trend into our measure of seasonal food security. The dotted blue line in **Fig 1B** shows the extrapolated average of seasonal food security in 2014 to 2015 and 2016 to 2017 in the sample, plus a linear time trend of 0.06 SD per year. Clearly, we could not associate all of the improved mental health from April 2020 to October 2020 to the onset of the pandemic; seasonal effects contributed to the improvement as well.

## The association between COVID-19 and mental health

The aggregate (cross-country) estimates of the association between the COVID-19 pandemic and mental health are shown in **Fig 2**. The 2 aggregate post-COVID-19 periods (0 to 4 months post-COVID-19 and >4 months post-COVID-19) are shown on the x-axis of each panel. The 3 panels of **Fig 2** vary in the set of samples that were included—the top-left panel includes all samples, the top-right panel includes only samples with either multiyear pre-COVID-19 waves or seasonal food security data, and the bottom-left panel includes only samples for which depression was measured with a complete validated mental health screening tool. In the top panels, we show results for the 3 methods of constructing the depression index for those samples without a full, validated scale: the factor analysis; the unweighted average (our preferred method); and the inverse-covariance weighted index.

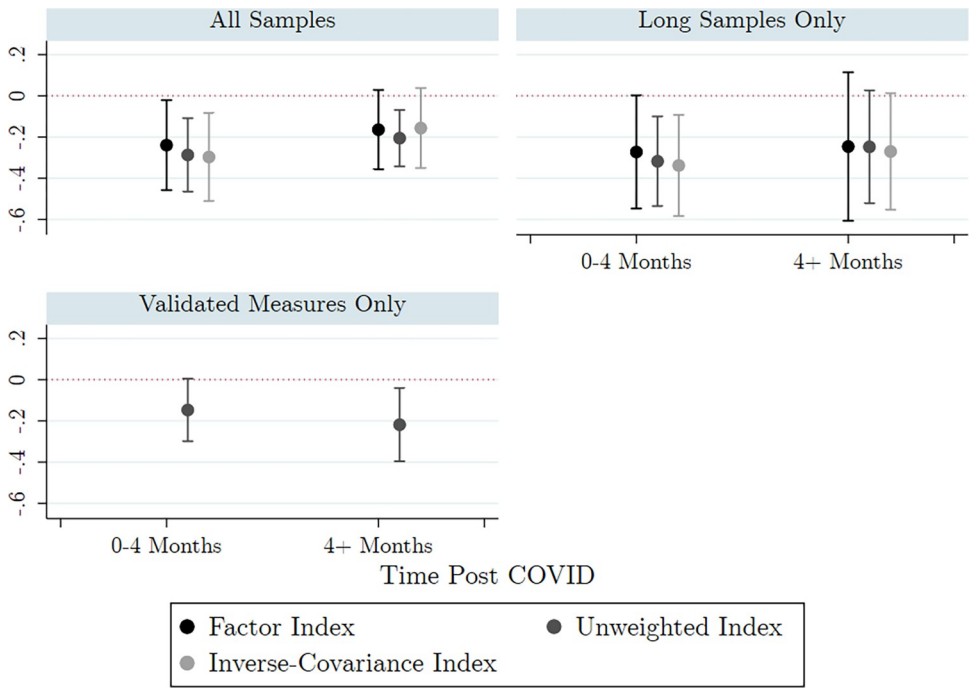

**Fig 2. Random effects aggregates different estimation strategies.**

The figure plots the average change in mental health from pre- to post-COVID-19 across samples. The y-axis is the SD change in mental health relative to pre-COVID. The x-axis shows the time after the onset of COVID-19 discretized into 2 groups: 0 to 4 months and more than 4 months. Point estimates are represented as dots with 95% CIs as whiskers. The color of the points indicates which method of weighting was used for our indices: the factor loadings (black, left), the unweighted index (dark gray, middle) or the inverse-covariance weights (light gray, right). The 3 panels are based on 3 different sets of samples. The first panel includes all samples. The second panel (top right) uses only samples with over a year of pre-COVID-19 mental health data (RWA, COL, KEN1) or supplemental data on typical seasonality in welfare (KEN2, NPL). The third panel (bottom left) uses only samples with validated depression measures (BGD, COL, NGA, DRC, KEN2). The average effect sizes and CIs were estimated from a random-effects model using restricted maximum likelihood to estimate the variance in true associations across samples. The estimates ranged from a minimum of −0.34 SD (the inverse-covariance weighted index, 0 to 4 months post-COVID-19, using samples with long panels or seasonality data) to a maximum of −0.08 SD (inverse-covariance weighted index, 4+ months post-COVID, samples with validated measures only).

Our short-run aggregates (0 to 4 months post-COVID-19) made use of the 6 samples where we had survey waves within 4 months after the onset of the pandemic: COL, KEN1, KEN2, KEN3, NPL, and RWA, further adjusted to the inclusion criteria per panel in **Fig 2**. The weights applied to these samples in the random-effects aggregate were 20, 13, 25, 15, 22, and 6, respectively, for our preferred depression measure. Weights were proportional to $1/((SE_s)^2 + \tau^2)$ where $SE_s$ was the standard error of the estimate in sample $s$, and $\tau^2$ was our estimate of the variance in "true association" across countries. RWA was weighted low due to its particularly large CIs. The longer-run aggregates made use of 9 samples with survey waves 4 months or more after the onset of the pandemic: BGD, COL, DRC, KEN1, KEN2, NGA, NPL, SLE, and

RWA. The longer-run aggregates weighed each sample more equally than the short-run aggregates at 13, 11, 13, 8, 12, 12, 13, 13, and 7, respectively, because the heterogeneity in most samples was larger 4+ months after the onset of the pandemic. We omitted one eligible sample from this analysis (KEN3) because—unlike the other samples—its long-run results were highly sensitive to model assumptions.

Overall, **Fig 2** shows high consistency in our aggregate results across methods of constructing the depression measure and across sample inclusion criteria in the 3 panels. In the short run, between 0 and 4 months after the onset of the pandemic, we saw a decline in depression of 0.24 SD (95% CI [−0.02,−0.46], $p = 0.03$) to 0.30 SD (95% CI [−0.47,−0.11], $p = 0.002$). In the longer run, correlational results showed that mental health rebounded somewhat. However, the improvement was limited, and the aggregate estimates remained negative, albeit not significantly different from zero. As a robustness check, **S3 Fig** repeated the analysis without adjusting for non-COVID-19 trends. This exercise produced qualitatively similar insights in the aggregate, but the coefficients were less precisely estimated. In our 5 samples with long pre-COVID-19 panels or seasonality data, for which we also had both short-term and longer-term estimates (COL, KEN1, RWA, KEN2, NPL), mental health improved on average by 0.07 SD (95% CI [−0.07, 0.22], $p = 0.41$) from 0 to 4 months to 4+ months.

**Table 1** shows the full results for each of the 10 samples separately. The dependent variable was the unweighted depression index. The results for the indices based on factor analysis and inverse-covariance weights in **S4** and **S5** **Tables,** respectively, were qualitatively similar. The table shows the samples where we adjusted for seasonal effects and time trends in the first 5 columns. Specifically, columns 1 to 3 are based on the full specification in Eq (1), and columns 4 and 5 are based on the models that control for seasonal food security as in Eq (2) and Eq (3)

**Table 1. Best estimates from each sample.**

| | Season + Time Trend | | | Seasonal Food Security Ctrl | | Time Control | Pre-Post Only | | | |
|---|---|---|---|---|---|---|---|---|---|---|
| | (1) RWA | (2) COL | (3) KEN1 | (4) KEN2 | (5) NPL | (6) KEN3 | (7) BGD | (8) NGA | (9) SLE | (10) DRC |
| 0–2 months | −0.251 (0.381) | −0.133 (0.0894) | −0.705 *** (0.193) | −0.249*** (0.0159) | −0.201*** (0.0617) | −0.0804 (0.0515) | | | | |
| 2–4 months | −0.429 (0.430) | | −0.873*** (0.183) | −0.201*** (0.0272) | −0.0623 (0.0627) | −0.199** (0.0930) | | | | |
| 4–6 months | −0.148 (0.309) | | −0.851*** (0.208) | −0.203*** (0.0405) | | | 0.0171 (0.0389) | | | |
| 6–9 months | 0.0118 (0.266) | −0.299*** (0.100) | | | 0.0580** (0.0292) | | | | | |
| 9–12 months | −0.268 (0.326) | | | | | | | | | |
| 12–15 months | | | | | | | | −0.356*** (0.0831) | −0.198*** (0.0342) | −0.241*** (0.0477) |
| Year | −0.284* (0.151) | 0.140** (0.0601) | 0.0172 (0.161) | | | 0.640*** (0.214) | | | | |
| Seasonal Food Security | | | | 0.454*** (0.106) | 0.134*** (0.0307) | | | | | |
| Observations | 1532 | 2503 | 5405 | 24,899 | 13,143 | 8342 | 6311 | 1081 | 6036 | 3133 |

Standard errors in parentheses.

*$p<.1$

**$p<.05$

***$p<.01$.

and Eq (4), respectively. The remaining 5 samples with less than 1 year of pre-COVID data and no supplemental data on food security trends are in columns 6 to 10.

The results were consistent with **Fig 2** for most samples. COL, KEN1, KEN2, NGA, and SLE showed negative and significant coefficients throughout the survey waves, implying a long-lasting correlation of COVID-19 and mental health. For RWA, the coefficients were negative and about −0.3 SD but not significant, largely due to increased noise in the estimation. NPL and DRC showed a reverse in sign over the longer run.

### Heterogeneity and decomposition of results

In the Supporting information, we included 2 additional analyses. First, in **S9, S10** and **S11 Tables**, we tested for heterogeneity in our estimates by gender, age, and socioeconomic status of respondents but found no systematic evidence of heterogeneity (only 4 out of 25 interaction terms were significant at the 5% level). Second, in **S4 Fig**, we decomposed the degree to which changes in mental health were associated with the stringency of COVID-19 policy measures and the number of COVID-19 cases, respectively. We could not reject that these 2 potential drivers of mental health problems equally contributed to our overall results. However, the coefficient estimate of a 1 SD change in policy stringency was roughly 3 times the magnitude of the estimate on a 1 SD change in (log) new COVID-19 cases. More details about the estimation method can be found in **S5 Appendix**.

### Alternative specification: Regression discontinuity

The RD estimates presented in **Table 2** for the NPL, COL, and KEN1 samples were similar to our estimates of the association 0 to 2 months post-pandemic in columns 2, 3, and 5 of **Table 1**. The RD estimate for KEN3 was effectively zero and insignificant. This is roughly in line with the relatively small coefficient in row 1, column (6) of **Table 1**. In contrast, the RD estimate for RWA was positive and significantly different from our estimate in **Table 1**. One reason for this difference was that our estimates in **Table 1** include month-of-year fixed effects, effectively comparing 2020 data to the same months in 2019. The months in which survey waves occurred post-COVID-19 in RWA were months with relatively high mental health outcomes in terms of seasonality, as deduced from pre-COVID-19 fluctuations. Another reason for the difference was that **Table 1** assumed pre-COVID-19 trends continued post-COVID-19

**Table 2. RD estimates from samples with surveys near pandemic onset.**

|  | (1) RE Aggregate | (2) COL | (3) KEN3 | (4) KEN1 | (5) NPL | (6) RWA |
|---|---|---|---|---|---|---|
| RD Estimate | −0.203 (0.166) | −0.124 (0.148) | −0.0678 (0.336) | −0.714*** (0.000) | −0.307*** (0.001) | 0.264* (0.094) |
| Observations |  | 2,503 | 6,680 | 5,405 | 12,601 | 1,159 |

*p*-Values in parentheses.

*p<.1

**p<.05

***p<.01.

Table shows regression discontinuity (RD) estimates of the effect for the onset of COVID lockdowns for samples COL (Colombia), KEN3 (Kenya 3), KEN1 (Kenya 1), NPL (Nepal), and RWA (Rwanda). RE stands for the aggregate estimate from the random-effects model. Dependent variable is the validated depression measure where available and the unweighted depression index where not. All measures are positively coded so that positive numbers indicate better mental health. The running variable is days prior to the onset of COVID restrictions. The date of restriction onset is set to the single highest daily increase in our COVID policy stringency index. See **S6 Table** for binned scatterplots and linear fits representing the RD design.

while the RD-specification estimated separate slopes for pre- and post-COVID-19 periods. This is easiest to visualize in the figure in **S6 Fig**, which shows binned scatterplots and linear fits from this RD design. Overall, the magnitude of the aggregate RD-estimate for these 5 countries is correlated with a decrease by around one-third and was no longer statistically significant ($p = 0.17$). The magnitude of our aggregate coefficient was robust to varying the bandwidths used in estimation, staying between 0.20 SD and 0.33 SD when we varied the time periods used from 30% to 100% of the available pre-period range in each sample **S5 Fig**.

## Discussion

We interviewed 21,162 individuals from 8 LMICs in Asia, SSA, and South America over multiple rounds both before and after the onset of the COVID-19 pandemic to assess their mental health status. We carefully estimated the correlation between the pandemic and mental health using multiple strategies to account for complications associated with seasonal fluctuations and time trends. Our analysis painted a clear, consistent, and robust picture: The pandemic was substantively and statistically significantly associated with worsened mental health in LMICs, especially during the early months of the pandemic. A random-effects aggregation across our samples showed that the index of depression symptoms correlated with an increase of 0.29 SD in the 4 months following the onset of the pandemic. This change in the depression score corresponded to a move from the 50th to the 63rd percentile in our median sample. Other estimates in the literature for LMIC populations find that a comprehensive anti-poverty intervention or asset transfers improve recipients' mental health by 0.14 to 0.16 SD [28,40]. The association in this paper was twice as large in the opposite direction.

Over the longer run, we saw some signs of a recovery, although aggregate mental health remained below pre-COVID-19 levels. Likewise, the longer-term estimates for most individual samples were consistently negative, reflecting a short-term reduction in mental health, which attenuated over time but did not fully bounce back to pre-pandemic levels. Two samples, however, showed positive coefficients for the longer-term period. For NPL, this likely reflected that mental health status returned to a pre-COVID-19 mean, as suggested in **Fig 1A**, since we could not separate the increase in mental health from normal seasonal dynamics. For DRC, similar explanations may hold but the interpretation is more speculative, as we lack data on seasonal variation in mental health.

We made an important methodological discovery that has implications for other COVID-19 mental health studies. We observed large within-year fluctuations in mental health indicators driven by agricultural crop cycles. Those fluctuations could dwarf the correlation between the onset of the pandemic and mental health, depending on exactly when COVID-19 had arrived in a country during the local crop cycle. A few other studies in agrarian settings have documented links between weather shocks, the time to harvest, and mental health outcomes such as well-being, cognitive function, and fluid intelligence [33,41,42]. This makes one-time, post-COVID-19 assessments of mental health outcomes difficult to interpret.

This insight has important implications for researchers and practitioners interested in tracking mental health in low-income settings. In agrarian societies, mental health fluctuates within the year, in line with periods of (pre-harvest) scarcity versus (post-harvest) plenty. Ignoring seasonal fluctuations can lead to spurious associations between other events (like a pandemic) and mental health, and bias estimates. Properly addressing the effects of seasonality requires multiple rounds of high-frequency data collected from a stable sample of individuals.

Our heterogeneity analyses showed that the decline in mental health was associated both with the stringency of the COVID-19 containment policies, as well as with the severity of the pandemic in terms of COVID-19 cases. However, the magnitude of the association was

roughly 3 times as large for the stringency measure as for the number of cases. This could have been due to the social and economic effects of lockdowns and other restrictions being more important to our respondents' mental health status than the direct health risks of the pandemic, or due to containment measures making the overall health risks of the pandemic more salient in a context where many respondents lacked accurate information on COVID-19 cases and evolving risks.

Existing multicountry studies of the mental health consequences of COVID-19 largely focus on HICs. Our study is a valuable addition to this literature because LMIC populations experienced the pandemic differently, faced more severe economic crises [1], and had worse access to health and economic support services. Our findings corroborate insights from rural Bangladesh and Colombia, which showed worsened maternal mental health during the pandemic [29,30]. They also supports findings from Bangladesh, demonstrating that food insecurity increased women's stress during the pandemic [7], and findings from Chile showing depression and anxiety symptoms increased among older adults since the onset of COVID-19 [43].

Our primary finding is consistent with the results from HIC studies, which mostly also found significant negative associations between COVID-19 and mental health. This includes a meta-analysis of 65 longitudinal cohort studies that compared mental health before versus during the COVID-19 pandemic (standardized mean change in depression 0.226 [95% CI: 0.109, 0.343]) [44], and a pre-post COVID-19 analysis that used helpline calls in HICs as a proxy for mental well-being [45]. Other individual country studies from Germany [14], United Kingdom (0.19 SD [95% CI: 0.17, 0.21]) [9,11,16], United States (0.27 SD (95% CI [0.23, 0.31], $p < 0.001$)) [10], and Canada [13] also found worse mental health during the COVID-19 pandemic. The effect size we estimated for LMICs in our meta-analysis is very similar to the estimates reported in the literature for HICs.

Our results stand in contrast to a Netherlands study that found no change in anxiety and depression symptoms [12], as well as a systematic review [15] that suggested no widespread negative impacts of COVID-19 on mental health (standardized mean difference [SMD] = 0.02, 95% CI −0.11 to 0.16). However, only one out of the 36 studies included in that review was conducted in a low-income setting, so the context of our data was notably different.

One potential limitation of our study is that the questions used to measure depression varied across surveys. However, previous research found a very high correlation between different screening measures of depression [46]. Additionally, brief depression screening instruments are typically as accurate as long ones in LMICs [47], and previous research has validated the use of a single-item depression measure [48]. Our results were generally consistent across surveys despite these differences in how depression was measured, suggesting that the findings in this paper are not dependent on one particular screening tool.

Another limitation is the small number of pre- and post-COVID-19 surveys available in most of our samples. Ideally, we would have had multiple rounds of pre-pandemic data for all samples to more fully account for seasonal effects and other time trends. Likewise, additional post-pandemic survey waves covering a longer time period would have allowed us to examine the post-COVID-19 dynamics and recovery processes in greater depth. This underscores the importance of collecting long-running panels in LMICs. Such data can greatly enhance countries' epidemic/pandemic preparedness, providing governments with crucial and timely information during health crises on the consequences of mitigation policies and the need for targeted support.

Relatedly, the RD analysis suffered from several limitations due the nature of the data we had: We lacked a density of observations just before the pandemic onset cutoff in several samples, and our running variable (survey date) had bunching at several points due to the discrete nature of survey waves. We included this analysis because it provided a different source of

variation (just before versus just after pandemic onset) than our estimates with individual fixed-effects and corrections for seasonality (comparing responses for the same individual and same months of the year). While each method suffered from its own potential biases, the relatively consistent set of results across methods raises confidence in our findings.

Finally, even though our study uses multiple datasets from 3 continents, our conclusions cannot necessarily be extrapolated to LMICs in general, given the enormous heterogeneity across countries that fall into this category. Further, 8 out of our 10 datasets are based on specific samples, and not nationally representative. Nevertheless, to our knowledge, it is the first and only existing study covering mental health across a diverse set of LMIC regions with pre- and post-COVID-19 panel waves, and the robustness of the findings across settings is reassuring. Future research may uncover additional datasets as mental health becomes a more popular topic of inquiry.

Mental health disorders are a large contributor to the global burden of disease, with severe repercussions for social welfare [49,50]. Poor people are often disproportionately affected, both because they are more frequently exposed to stressful events, and because they have more limited access to treatment. COVID-19 has exacerbated mental health issues and exposed the weaknesses of existing health systems [51,52]. Our results suggest that it has become important to devise plans to protect mental health, because the pandemic may leave behind a persistent depression that affects long-run health and productivity even after the virus subsides.

LMICs' health systems are deficient in terms of resources allocated to mental health, the required workforce, and availability of essential psychotropics [24]. Moreover, there is greater social stigma around mental health issues in those settings [22]. Given the dearth of trained mental healthcare providers in LMICs, innovative alternative solutions are required. Recent evidence shows that psychotherapy delivered by nonspecialists is a cost-effective treatment for depression even after 5 years [53,54]. Considering the enormous returns from engaging nonspecialists in this sector, policymakers could prioritize task-sharing in mental healthcare. Frugal solutions like the training of paraprofessionals, lay health workers, and community-based care providers would be a great source of support for increasing the accessibility and affordability of mental health services. Such support can be included in universal healthcare coverage [54].

In sum, we document a large, significant, negative association between the pandemic and mental health, especially during the early months of the pandemic when COVID-19 mitigation measures were most stringent. Absent policy interventions, the pandemic could result in long-lasting depression, especially in LMICs. Our data analysis also identified strong seasonal patterns in mental health outcomes in low-income settings associated with seasonality in agricultural incomes. Policy designs and their evaluations should be cognizant of such seasonal variations in mental well-being.

## Supporting information

**S1 Fig. Data collection and lean season timeline.** Figure plots the data collection and lean season timeline across samples to show which crop cycle the COVID-19 arrived in the countries. The y-axis shows the sample. The x-axis shows the calendar time. Dark blue color refers to individual surveys during the data collection, while dark orange color shows the main lean season across countries and years. The red vertical line shows the onset of the COVID-19 pandemic on 11 March 2020. Data are retrieved from the FAO Global Information and Early Warning System. For further information about definitions, please see the FAO's website. (PDF)

**S2 Fig. Mental health, lockdowns, and cases.** (PDF)

**S3 Fig. Random effects aggregates with individual fixed effects and no additional controls.** Figure plots the average change in mental health pre-post COVID across samples using a common estimation strategy that uses individual fixed-effects without additional controls for all samples. The y-axis is the change in mental health relative to pre-COVID. The x-axis shows time after the onset of COVID discretized into 2 groups: 0 to 4 months and more than 4 months. Point estimates are represented as dots with 95% confidence intervals as whiskers. The color of the points indicates which method of weighting our indices was used: the factor loadings (black, left), the unweighted index (dark gray, middle), or the inverse-covariance weights (light gray, right). The average effect sizes and confidence intervals are estimated from a random-effects model using restricted maximum likelihood to estimate the variance in true effects across samples.
(PDF)

**S4 Fig. Relationship between COVID-19 policies, cases, and depression.** Figure shows estimates of the relationship between unweighted depression index and COVID-19 policies and cases.
(PDF)

**S5 Fig. Sensitivity of RD estimates to time duration used in estimation.** Figure shows regression discontinuity estimates of change in depression following onset of the pandemic with varying bandwidths used in estimation. The x-axis shows the share of the available pre-COVID time period used in each sample used in estimation. In each estimate, the sample was reduced so that only the final **D** days of data prior to and after the onset of the pandemic were used, where **D** was set to be a certain percentage of the range of the data prior to the onset of the pandemic. We aggregate the results for each percentage of the available time used using the same random-effects model used in **Fig 2**.
(PDF)

**S6 Fig. Regression discontinuity plots for 5 samples.** Figure shows a graphical representation of the regression discontinuity (RD) estimates for each of the 5 samples in Table 2. The x-axis is the running variable in the RD design, days post onset of lockdowns in that country. The y-axis is standard deviation units of our unweighted depression index. The points are the average values of the index within discrete time bins. The solid lines are linear fits to the points shown pre and post the onset of the lockdown.
(PDF)

**S1 Appendix. Sample description.**
(PDF)

**S2 Appendix. The STROBE Checklist.**
(PDF)

**S3 Appendix. Analysis plan.**
(PDF)

**S4 Appendix. Inclusivity in global research.**
(PDF)

**S5 Appendix. Relationship between mental health, COVID cases, and COVID restrictions.**
(PDF)

**S1 Table. Description of household survey data samples used in the analysis.**
(PDF)

**S2 Table. Descriptive statistics.**
(PDF)

**S3 Table. Question wording and answer options for depression measurement.** This table presents question wording and answer options for mental health variables used in analysis. In BGD, COL, DRC, KEN3, and NGA samples, we followed the prescribed calculation method of each scale. In the remaining samples, we constructed indices of the available depression items as explained in the Data and Methods section. In BGD and KEN3, the total CES-D score (range: 0 to 80 and 0 to 30, respectively) is calculated by summing all the items. In COL, the SCL-90 score is calculated by dividing the raw score to number of questions in the depression domain ($n = 13$). In DRC, the final score is calculated by dividing the sum of the scores of all the items by 25 (the final score ranges from 1.00 to 4.00), where at least 22 items had to be answered for the assessment to be considered valid. In NGA, the total raw score, ranging from 0 to 25, is multiplied by 4 to give the final score, with 0 representing worst possible, whereas a score of 100 representing best possible quality of life. All scores are reverse coded and standardized by mean = 0, standard deviation = 1. The higher scores represents better mental well-being for each scale.
(PDF)

**S4 Table. Best estimates from each sample (ICW Index).**
(PDF)

**S5 Table. Best estimates from each sample (Factor Index).**
(PDF)

**S6 Table. Pre-post differences in depression index.**
(PDF)

**S7 Table. Pre-post differences in depression index (Inverse-covariance weights).**
(PDF)

**S8 Table. Pre-post differences in depression index (Unweighted index).**
(PDF)

**S9 Table. Heterogeneity in estimates by gender.**
(PDF)

**S10 Table. Heterogeneity in estimates by socioeconomic status.**
(PDF)

**S11 Table. Heterogeneity in estimates by age.**
(PDF)

## Acknowledgments

We are indebted to study participants for generously giving their time. We are grateful to the staff of Yale Research Initiative on Innovation and Scale (Y-RISE); Centre for the Study of Labour and Mobility in Nepal (CESLAM); The Amsterdam Institute for Global Health and Development (AIGHD), Richard de Groot, Menno Pradhan; Africa Population and Health Research Center (APHRC), Estelle Sidze, Caroline Wainaina; 100WEEKS Director Jeroen de Lange and Rwanda Country Director Gervais Nkurunziza; Semillas de Apego team: Arturo

Harker, Blasina Niño, Vilma Reyes, Alicia Lieberman, María José Torres and Juliana Sánchez; IPA Nigeria's Country Director, Emeka Eluemunor and IPA's Regional Director for West Africa, Claudia Casarotto; the Sierra Leone Ministry of Energy and Madison Levine, Joseph Levine, Vasudha Ramakrishna; Sarah Khan, Morgan Holmes, the late Jean Paul Zibika, Amani Matabaro Tom and the team at Forcier Consulting, Search for Common Ground, the IMA World Health and funded by the USAID; Innovations for Poverty Action Kenya, Eric Ochieng, Ronald Malaki, Michelle Layvant, Somara Sobharwal, Pooja Suri and Eve Zhang; Vyxer Remit Kenya, Carol Nekesa, Andrew Wabwire, Layna Lowe, Anya Marchenko, Gwyneth Miner, Carlos Paramo, Tilman Graff and Magdalena Larreboure; BRAC Institute of Governance and Development, BRAC Institute of Educational Development, Monash University Australia.

## Author Contributions

**Conceptualization:** Nursena Aksunger, Corey Vernot, Wendy Janssens, Ahmed Mushfiq Mobarak.

**Data curation:** Corey Vernot, Maarten Voors, Niccolò F. Meriggi, Katherine Dai.

**Formal analysis:** Nursena Aksunger, Corey Vernot.

**Funding acquisition:** Maarten Voors, Niccolò F. Meriggi, Wendy Janssens, Ahmed Mushfiq Mobarak.

**Methodology:** Nursena Aksunger, Corey Vernot, Rebecca Littman, Wendy Janssens, Ahmed Mushfiq Mobarak.

**Visualization:** Corey Vernot, Ahmed Mushfiq Mobarak.

**Writing – original draft:** Nursena Aksunger, Corey Vernot, Rebecca Littman, Maarten Voors, Niccolò F. Meriggi, Wendy Janssens, Ahmed Mushfiq Mobarak.

**Writing – review & editing:** Nursena Aksunger, Corey Vernot, Rebecca Littman, Maarten Voors, Niccolò F. Meriggi, Amanuel Abajobir, Bernd Beber, Dennis Egger, Asad Islam, Jocelyn Kelly, Arjun Kharel, Amani Matabaro, Andrés Moya, Pheliciah Mwachofi, Carolyn Nekesa, Eric Ochieng, Tabassum Rahman, Alexandra Scacco, Yvonne van Dalen, Michael Walker, Wendy Janssens, Ahmed Mushfiq Mobarak.

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
