## [Editor Report · Decision Letter 0]

26 Jul 2022

Dear Dr Mobarak, 

Thank you for submitting your manuscript entitled "The Effect of the COVID-19 Pandemic on Mental Health in Low and Middle Income Countries" for consideration by PLOS Medicine.

Your manuscript has now been evaluated by the PLOS Medicine editorial staff and I am writing to let you know that we would like to send your submission out for external peer review.

Your manuscript is currently under consideration as part of the Special Issue on the COVID-19 pandemic and global mental health. The deadline for the Special Issue is being extended to December 15 2022, with anticipated publication in Q1 2023 (subject to change dependent on submission volume). We intend to publish all papers accepted for the Special Issue simultaneously.

Given that this extension was announced after you submitted your manuscript for consideration, we appreciate that you may no longer wish for your manuscript to considered specifically for the Special Issue. If this is the case, or if you have any other questions, please feel free to contact me (cdavidson@plos.org) and this can be discussed.

Before we can send your manuscript to reviewers, we need you to complete your submission by providing the metadata that is required for full assessment. To this end, please login to Editorial Manager where you will find the paper in the 'Submissions Needing Revisions' folder on your homepage. Please click 'Revise Submission' from the Action Links and complete all additional questions in the submission questionnaire.

Please re-submit your manuscript within two working days, i.e. by Jul 28 2022 11:59PM.

Kind regards,

Callam Davidson

Associate Editor

PLOS Medicine

---

## [Decision Letter · Decision Letter 1]

28 Sep 2022

Dear Dr. Mobarak,

Thank you very much for submitting your manuscript "The Effect of the COVID-19 Pandemic on Mental Health in Low and Middle Income Countries" (PMEDICINE-D-22-02441R1) for consideration for the upcoming special issue on the pandemic and global mental health at PLOS Medicine. 

Your paper was evaluated by an associate editor and discussed among all the editors here. It was also discussed with the guest editors of the special issue, and sent to independent reviewers, including a statistical reviewer. The reviews are appended at the bottom of this email and any accompanying reviewer attachments can be seen via the link below:

[LINK]

In light of these reviews, I am afraid that we will not be able to accept the manuscript for publication in the journal in its current form, but we would like to consider a revised version that addresses the reviewers' and editors' comments. Obviously we cannot make any decision about publication until we have seen the revised manuscript and your response, and we plan to seek re-review by one or more of the reviewers. 

We hope to receive your revised manuscript by Oct 19 2022 11:59PM. Please email us (plosmedicine@plos.org) if you have any questions or concerns.

We look forward to receiving your revised manuscript. 

Sincerely,

Callam Davidson, 

PLOS Medicine

plosmedicine.org

Substantial editing is required to align the manuscript with PLOS Medicine guidelines. In addition to the comments below, please review the instructions at https://journals.plos.org/plosmedicine/s/submission-guidelines and ensure your manuscript meets the requirements. 

Please revise your title according to PLOS Medicine's style. Your title must be nondeclarative and not a question. It should begin with main concept if possible. "Effect of" should be used only if causality can be inferred, i.e., for an RCT. Please place the study design ("A randomized controlled trial," "A retrospective study," "A modelling study," etc.) in the subtitle (ie, after a colon).

Please structure your abstract using the PLOS Medicine headings (Background, Methods and Findings, Conclusions).

Abstract Background: Provide the context of why the study is important. The final sentence should clearly state the study question.

Abstract Methods and Findings:

* Please ensure that all numbers presented in the abstract are present and identical to numbers presented in the main manuscript text.

* Please include the study design, population and setting, number of participants, years during which the study took place, length of follow up, and main outcome measures.

* Please quantify the main results (with 95% CIs and p values).

* Please include the important dependent variables that are adjusted for in the analyses.

Abstract Conclusions:

* Please address the study implications without overreaching what can be concluded from the data; the phrase "In this study, we observed ..." may be useful.

* Please interpret the study based on the results presented in the abstract, emphasizing what is new without overstating your conclusions.

* Please avoid vague statements such as "these results have major implications for policy/clinical care". Mention only specific implications substantiated by the results.

The Data Availability Statement (DAS) requires revision. For each data source used in your study: 

Please justify your response ‘My study does not require an ethics statement’ in the submission form. This does not appear to be correct.

We note that you conducted research or obtained samples in a foreign country. Did you consider including a local author as first or last author? If not, we recommend that you consider doing so in line with ICMJE's authorship requirements (https://www.icmje.org/recommendations/browse/roles-and-responsibilities/defining-the-role-of-authors-and-contributors.html). PLOS has a parachute research policy which aims to promote collaboration and inclusivity in global health research. You are required to complete PLOS’ questionnaire on inclusivity in global research and submit it with your revised paper. The policy and questionnaire can be found at https://journals.plos.org/plosone/s/best-practices-in-research-reporting.

Please include continuous line numbering throughout your manuscript. 

Citations should be in square brackets, and preceding punctuation. Please use the "Vancouver" style for reference formatting, and see our website for other reference guidelines https://journals.plos.org/plosmedicine/s/submission-guidelines#loc-references

Please remove the study results and/or conclusion from the Introduction.

Please conclude the Introduction with a clear description of the study question or hypothesis.

Your manuscript needs to be organised using the PLOS Medicine submission guidelines. Please see https://journals.plos.org/plosmedicine/s/submission-guidelines and organise your manuscript per the instructions found under ‘Manuscript Organisation’.

Please label, cite, and present your Supporting Information as outlined here: https://journals.plos.org/plosmedicine/s/supporting-information

Much of what is presented in the Methods and, to some extent, the Results section (in terms of e.g., the relative merits and weaknesses of the different methodologies applied) would be better placed in the Discussion. Please use the Methods section to clearly describe the study conduct and analyses, in sufficient detail to allow other researchers to replicate your study. Please use the Results to describe your data without including interpretation/questions for the reader (this can be saved for the Discussion). Again, please review our submission guidelines for more information on PLOS Medicine style: https://journals.plos.org/plosmedicine/s/submission-guidelines.

Your study is observational and therefore causality cannot be inferred. Please remove language that implies causality, such as ‘Effects’. Refer to associations instead.

Footnotes are not permitted. Please move the information into the main text or the reference list, depending on the content.

Please rename your ‘Conclusion and Implications for Policy’ section ‘Discussion’.

Please present and organize the Discussion as follows: a short, clear summary of the article's findings; what the study adds to existing research and where and why the results may differ from previous research; strengths and limitations of the study; implications and next steps for research, clinical practice, and/or public policy; one-paragraph conclusion.

Please remove the List of Abbreviations.

Your Acknowledgements section contains details of the Ethical approval for the study – this information should be relocated to the Methods, alongside details of how informed consent was sought from participants (including whether this was written or verbal). 

Please remove the ‘Authorship’ section as this is captured as metadata based on the Submission Form responses. 

Please ensure that the study is reported according to the STROBE guideline, and include the completed STROBE checklist as Supporting Information. Please add the following statement, or similar, to the Methods: "This study is reported as per the Strengthening the Reporting of Observational Studies in Epidemiology (STROBE) guideline (S1 Checklist)."

Did your study have a prospective protocol or analysis plan? Please state this (either way) early in the Methods section.

For all observational studies, in the manuscript text, please indicate: (1) the specific hypotheses you intended to test, (2) the analytical methods by which you planned to test them, (3) the analyses you actually performed, and (4) when reported analyses differ from those that were planned, transparent explanations for differences that affect the reliability of the study's results. If a reported analysis was performed based on an interesting but unanticipated pattern in the data, please be clear that the analysis was data-driven.

Comments from the reviewers:

Reviewer #1: Excellent analysis and presentation. However, the document must expose the possible limitations of the study, for example, the diversity of instruments used to quantify depressive symptoms. These instruments have a substantial impact on the frequency of reported symptoms.

Reviewer #2: Thanks for the opportunity to review your manuscript. My role is as a statistical reviewer, so my review concentrates on the study design, data, and analysis that are presented. I have put general questions first, followed by queries relevant to a specific section of the manuscript (with a page/paragraph reference).

This study aims to quantify changes in levels of depression in people from LMICs associated with the COVID-19 pandemic. Data is aggregated from existing research in LMICs where measures of depression are collected. Levels of depression are taken from both validated and non-validated measures. Validated measures are standardised (to units of SD), un-validated ones have each relevant item standardised, then several different strategies to combine these items together (with and without weights). Each study has different periods of data collection - some studies allow for direct assessment of seasonal and underlying trend. In each study, a measure of the stringency of COVID-19 public health interventions is derived. A multilevel model (to incorporate repeated measurements of the same individuals over time) is used in each sub-sample to estimate difference between COVID and non-COVID periods (indicator variable), with effect of year and month (via binary indicators) also included. The estimate of difference between COVID and non-COVID periods is then included in a random-effects meta-analysis to provide an overall estimate of the impact of the COVID period on the depression score across all the sub-studies. Sensitivity analyses considering only sub-samples with a longer-times series, or restricted to the validated measurements of depression are considered. An interrupted time-series for sub-samples is also considered. The manuscript includes detailed and well-organised supplementary material, thank you, this makes reviewing much easier. 

To my (non-expert) eyes, the questions in some of the depression measurements seem to be very different (Table A2). E.g. some of these concentrate on symptoms of depression (COL study), while in others it is effectively a self-report question rather than a depression scale (KEN1). My concern is whether some of these should be included in the study if there is a so much difference between them (sensitivity analysis non-withstanding). 

The study and conclusions are described as generalizable to LMICs, while the data mostly comes from ad-hoc surveys in LMICs. Only 2/10 underlying samples have a population-based sampling frame, the rest are from specific samples within each country. The studies are based in countries from South Asia, Africa, and one South American country, and I think there are too many regions missed to be able to say that the overall study represents LMICs broadly. 

In the main analyses and the ITS (discontinuity), what checks were made of the models, e.g. form of residuals, checks for influential observations, autocorrelation? 

Much of the language used to describe the results e.g. 'the pandemic produced substantively and statistically significant increase in depression in LMICs' is causal. I would consider tempering the expression given that time-series are vulnerable to confounding, and detail some of the limitations of these methods in the discussion.

P2, Paragraph 1. A change in units of standard deviation is difficult to interpret. Is there any context that could be added to the abstract (and elsewhere) to indicate what this translates in practice? 

P3, Paragraph 3. I would typically expect the introduction to introduce the context and research question rather than include results. Similarly, there is parts of the methodology in the introduction (P4).

P7, Paragraph 1. How is the 'lean' season in each of the sub-studies defined? 

P8, Paragraph 8. To clarify, is each of the nine indicators a binary variable? i.e. there are 9 possible values for the index to take? 

P9, Paragraph 5. What covariance structure was used for the repeated measurements of the same participants? 

Were survey weights from the population-based studies used in these analyses? 

P10, Paragraph 2. Will this approach detect a secular trend in depression occurring within 1 year, or will that all be absorbed into the fixed effects for month across all the years of data? 

P11, Paragraph 4. What software was used for the analyses? 

P17, Paragraph 1. Much of this material should be moved to a method section. 

Was a sufficient number of time-points (e.g. 8 is often suggested) of data available before and after lockdown in each study? 

P18, Table 2. The trend line in these analyses extends beyond available data - e.g. with only a short period of data from KEN1 the post-COVID prediction line extends out 10 times the extend of available post-change data. These should be revised to the prediction line ends when there is no more data. 

In the post-onset time-period, there is also a line fitted from only two points in time for the Colombia_NW sub-sample. I understand that these points are averages of the sample at each time, but there are serious risks to trying to infer a slope change from only two time-points even if there are many measurements at each time. 

Reviewer #3: This paper presents data from ten longitudinal pre-/post COVID-19 data sets spanning eight countries to examine changes in mental health from upward of 21,000 people in Low and Middle Income Countries (LMICs). Using creative and careful modelling, the authors combine impressive data from understudied regions to demonstrate the significantly negative change in mental health in the first few (~4) months of the pandemic, which reduces somewhat overtime. This trend is most noticeable when accounting for the strong seasonal variation in mental health that is shaped by food production patterns. 

I thought there was much to be appreciated in this work. First, this paper provides an extremely important contribution to the literature by shedding light on the mental health changes in LIMCs during COVID-19 onset. As the authors note, we have learned much about mental health fluctuations during the pandemic in wealthy nations but the data are sparse elsewhere. Thus this fills a dire hole in our understanding of mental health under the COVID-19 pandemic. Second, I appreciated the thoughtful aggregation of the data and careful analytic strategy that modelled each country with the best information available, but also considered a generic approach for consistency. Third, as the authors make clear, attention paid to the local context and season patterns is critical to understand mental health changes. 

Results seem clear and consistent with most of the findings I have read to date, and meta-analyses in the literature. Depression levels peaked soon after COVID-19 onset and began to return (although not fully cover) to pre-pandemic levels in most countries several months later. 

My questions primarily have to do with whether greater precision can be inferred from this impressive database:

1. The authors speak to mental health changes from pre- to post-COVID-19 but is it possible to distinguish this further? For instance, are respondents reporting higher levels of depression in response to higher local case counts, higher deaths, greater policy stringency, or disruption from daily life? The authors briefly mention the importance of these alternative factors but I don't see them included in the models. 

2. The authors demonstrate the importance of appropriately capturing time/season variations in agricultural food cycles that impact food security at-scale. I wondered: Do these findings parallel at the individual level, such that people with lower incomes/greater food or work precarity report larger variations in mental health responses? More broadly, what heterogeneity is seen in the data? Are parents, elderly, the young, females/males at greater risk for depression, perhaps due to greater vulnerability?

3. I notice in the appendix that some countries provided various social protection efforts, such as cash transfers to various industries. Could this provide a within-country robustness check, wherein recipients of such transfers or bursaries report better mental health than would be expected otherwise?

4. Are the authors able and willing to benchmark changes in mental health observed in the LIMC to those observed elsewhere in higher income countries? It could be helpful to know if the observed increase in depression was larger in size in LIMCs to potentially garner greater aid. 

5. Along similar lines, I found the recommendations offered at the very end of the paper to be brief and broad. While training a fleet of paraprofessionals may be one strategy, I wondered whether any other more targeted suggestions could be provided based uniquely on the current data re: scope and timing of threat, and/or heterogeneity in response.

[LINK]

---

## [Decision Letter · Decision Letter 2]

7 Dec 2022

Dear Dr. Mobarak,

Thank you very much for re-submitting your manuscript "COVID-19 and mental health: A prospective cohort study in eight low-and middle-income countries" (PMEDICINE-D-22-02441R2) for review by PLOS Medicine.

I have discussed the paper with my colleagues and the academic editor and it was also seen again by one reviewer. I am pleased to say that provided the remaining editorial and production issues are dealt with we are planning to accept the paper for publication in the journal.

[LINK]

We look forward to receiving the revised manuscript by Dec 14 2022 11:59PM.   

Sincerely,

Callam Davidson, 

Associate Editor 

PLOS Medicine

plosmedicine.org

Requests from Editors:

Please update your title to ‘COVID-19 and mental health in eight Low- and Middle-Income Countries: A prospective cohort study’.

Please ensure your Methods and Results are consistently reported in the past tense (currently the manuscript is a mixture of past and present).

Abstract Conclusions: I feel that leading with the conclusions relating to agriculture has the potential to confuse readers, as the seasonal component does not feature prominently. I would recommend leading with the second sentence and including the information regarding crop cycles nearer the end of the subsection.

The DOI in your data availability statement currently leads to an error page – please confirm this will be updated on publication?

I noted issues with Figure S1 (duplicated content in the legend), Figure S5 (‘??’ in the legend), and Figure S2 (not enough detail in the legend to allow readers to interpret figure in its own right; additionally, panels are too small to read and need to be enlarged. 

Thank you for the extensive work you have done to restructure your manuscript in line with PLOS Medicine guidelines, this is much appreciated. I still feel that some of the content in the Methods section would be better placed in the Results (for example, lines 136-138) or the Discussion (for example, lines 269-278). The Methods should only report what was done in sufficient detail to allow for reproduction of the study.

Related to the above, the Results also contain information that would be better placed in the Discussion (e.g., lines 333-336; lines 436-440). Authors should avoid speculation/interpretation in this section and instead aim to objectively report what was observed.

Authors should avoid making causal assertions given the observational design (refer instead to suggestions/associations). Examples are apparent at lines 348-350, line 401, line 464, lines 467-468, line 475, line 480, and line 566, but please carefully check throughout. 

The terms gender and sex are not interchangeable (as discussed in https://www.who.int/health-topics/gender); please ensure you are using the appropriate term throughout.

References 8 and 15 need [preprint] adding (see https://journals.plos.org/plosmedicine/s/submission-guidelines#loc-references).

Please ensure any references that include internet sources contain the date cited (see https://journals.plos.org/plosmedicine/s/submission-guidelines#loc-references).

Figure 1: Please label the y-axes.

Table 2: The embedded Figures are too small to read – I would propose separating these out into a separate figure. 

Table 2 legend: Please define abbreviations used in the Table.

Comments from Reviewers:

Reviewer #2: Thanks for the revised manuscript and responses to my review. I am happy with the changes to the manuscript which have resolved the queries from my initial review. The updates to the analysis plan as the study proceeded look like reasonable decisions to me. The explanation about the interpretation of the effect size in SD units is good from my (statistical) perspective. I think the discussion about the limitations is fair and clear around issues about causality and generalisability.

[LINK]

---

## [Editor Report · Decision Letter 3]

3 Jan 2023

Dear Dr. Mobarak,

Thank you very much for re-submitting your manuscript "COVID-19 and mental health in eight Low- and Middle-Income Countries: A prospective cohort study" (PMEDICINE-D-22-02441R3) for review by PLOS Medicine.

Before we can accept the paper for publication in the journal, we require that you address a few remaining issues.

[LINK]

We look forward to receiving the revised manuscript by 10 January 2023.   

Sincerely,

Callam Davidson, 

Associate Editor 

PLOS Medicine

plosmedicine.org

Requests from Editors:

Please shorten your Author Summary by combining bullet points one and two as follows - 'The worldwide economic and health crises triggered by the COVID-19 pandemic have had a significant influence on mental health, with the estimated prevalence of depression having increased by more than 25 percent in high-income countries'.

Please relocate the first sentence from bullet point four such that it forms part of bullet point three - 'Although the adverse consequences of the pandemic on living standards have been most severe in low- and middle-income countries (LMICs), the consequences of the pandemic for mental health in LMICs have received less attention'.

The remainder of bullet point four ('This study was intended to determine the relationship between the COVID-19 outbreak and mental health in eight LMICs') can be deleted as this information is included in bullet point five. 

Please correct 'epidemic' to 'pandemic' in bullet point nine. 

Please correct 'impact' to 'association' in bullet point ten.

Please aim to condense the bullet points under the question 'What do these findings mean?' such that there are three rather than four bullets, and avoid directly recycling language from the Abstract where possible.

Please update 'impact of the pandemic on mental health' to 'association of the pandemic with mental health' to reflect the observational nature of the research. 

Please check throughout for other instances of causal language (I noted several other instances of the term 'impact').

---

## [Editor Report · Decision Letter 4]

12 Jan 2023

Dear Dr Mobarak, 

On behalf of my colleagues and the Academic Editor, Professor Vikram Patel, I am pleased to inform you that we have agreed to publish your manuscript "COVID-19 and mental health in eight Low- and Middle-Income Countries: A prospective cohort study" (PMEDICINE-D-22-02441R4) in PLOS Medicine.

When making the formatting changes, please also address the following editorial comments:

* Author Summary, bullet point 6: please correct 'is' to 'were'.

PRESS

Thank you again for submitting to PLOS Medicine. We look forward to publishing your paper and will be in touch soon with further information regarding the planned publication window for the Special Issue.

Sincerely, 

Callam Davidson 

Associate Editor 

PLOS Medicine